



# Mathematical foundation of Capon's method for planetary magnetic field analysis

Simon Toepfer[1], Yasuhito Narita[2,3], Daniel Heyner[3], Patrick Kolhey[3], and Uwe Motschmann[1,4]

[1]Institut für Theoretische Physik, Technische Universität Braunschweig, Braunschweig, Germany
[2]Space Research Institute, Austrian Academy of Sciences, Graz, Austria
[3]Institut für Geophysik und extraterrestrische Physik, Technische Universität Braunschweig, Braunschweig, Germany
[4]DLR Institute of Planetary Research, Berlin, Germany

**Correspondence:** Simon Toepfer (s.toepfer@tu-braunschweig.de)

**Abstract.**

Minimum variance distortionless projection, the so-called Capon method, serves as a powerful and robust data analysis tool when working on various kinds of ill-posed inverse problems. The method has not only successfully been applied to multi-point wave and turbulence studies in the context of space plasma physics, but also is currently being considered as a technique to

perform the multipole expansion of planetary magnetic fields from a limited data set, such as Mercury's magnetic field analysis. The practical application and limits of the Capon method are discussed in a rigorous fashion by formulating its linear-algebraic derivation in view of planetary magnetic field studies. Furthermore, the optimization of Capon's method by making use of diagonal loading is considered.

## 1 Introduction

Nonlinear and adaptive filter technique has a wider range of applications in geophysical and space science studies to find the most likely parameter set describing the measurement data or to decompose the data into a set of signals and noise. Above all, the minimum variance distortionless projection introduced by Capon (1969) (hereafter, Capon's method) has successfully been applied to multi-point data analyses for the waves, turbulence fields, and current sheets (Motschmann et al., 1996; Glassmeier et al., 2001; Narita et al., 2003, 2013; Contantinescu et al., 2006; Plaschke et al., 2008). The strength of Capon's method

lies in the fact that the method performs a robust data fitting even when the spatial sampling or data amount is limited in the measurement (e.g., successfully applied to four-spacecraft data (Motschmann et al., 1996)). Capon's method is currently being considered for planetary magnetic field studies in which the data (i.e., magnetic field samples) are more limited (e.g., non-ideal orbits), and has recently been applied to Mercury's magnetic field data in view of the BepiColombo mission (Toepfer et al., 2020).


From a theoretical point of view there exist several origins for the derivation of the method. The first derivation of Capon's method (Capon, 1969), constructed for the analysis of seismic waves, is based on the estimation of frequency-wavenumber spectra. Later on, this approach has been reformulated in terms of matrix algebra (Motschmann et al., 1996). In the light





of mathematical statistics Capon's estimator can be regarded as a special case of the maximum likelihood estimator (Narita, 2019). In this work the linear-algebraic formulation of the method (Motschmann et al., 1996) with specific attention to the magnetic field analysis is extended and the application of diagonal loading is discussed to improve the quality of data analysis with more justified applications and limits.

## 2 Motivation of Capon's method

The analysis of planetary magnetic fields is of great interest and one of the main tasks in space science. Here we pay special attention to the analysis of Mercury's internal magnetic field which is one of the primary goals of the BepiColombo mission (Benkhoff et al., 2010). The magnetometer on board the Mercury Planetary Orbiter (MPO) (Glassmeier et al., 2010) measures the magnetic field vectors $\boldsymbol{b}^i = \left(b_x^i, b_y^i, b_z^i\right)^T \in \mathbb{R}^3$ at $N$ data points $\boldsymbol{x}^i$, $i = 1, ..., N$ along the orbit in the vicinity of Mercury. The magnetic fields around Mercury are considered as a composition or superposition of internal fields generated by the dynamo process, crustal and induced fields, which are mainly dominated by dipole and quadrupole fields and external fields generated by the currents flowing in the magnetosphere. For Mercury the external fields contribute a significant amount to the total magnetic field within the magnetosphere (Anderson et al., 2011) and therefore, a robust method is required for seperating the internal fields from the total measured field. Yet, each component has to be properly modeled and parameterized when decomposing the field.

For example, when only data in current-free regions are analyzed, the planetary magnetic field is irrotational and can be parameterized via the Gauss representation (Gauss, 1839). Whithin these current-free regions, the internal magnetic field can be expressed as the gradient of a scalar potential $\Phi$, which can be expanded into a set of basis functions. Considering the expansion into spherical harmonics, the potential for the planetary dipole and quadrupole fields results in

$$\Phi(\boldsymbol{x}^i) = R_{\mathrm{M}} \sum_{l=1}^{2} \left(\frac{R_{\mathrm{M}}}{r^i}\right)^{l+1} \sum_{m=0}^{l} \left[g_l^m \cos(m\lambda^i) + h_l^m \sin(m\lambda^i)\right] P_l^m\left(\cos(\theta^i)\right), \tag{1}$$

where planetary centered coordinates with radius $r^i \in [R_{\mathrm{M}}, \infty)$, azimuth angle $\lambda^i \in [0, 2\pi]$ and polar angle $\theta^i \in [0, \pi]$ are chosen. $R_{\mathrm{M}}$ indicates the radius of Mercury and $P_l^m$ are the Schmidt-normalized associated Legendre polynomials of degree $l$ and order $m$. Within this series expansion there occurs a set of expansion coefficients $g_l^m$ and $h_l^m$, named internal Gauss coefficients. By constructing the Gauss coefficients in a vectorial fashion and defining the true coefficient vector as $\boldsymbol{g} = \left(g_1^0, g_1^1, h_1^1, g_2^0, g_2^1, h_2^1, g_2^2, h_2^2\right)^T$, the magnetic field vectors at every data point $\boldsymbol{x}^i$ and the expansion coefficients are related via

$$\boldsymbol{b}^i = -\partial_{\boldsymbol{x}^i} \Phi(\boldsymbol{x}^i) + \boldsymbol{v}^i + \boldsymbol{n}^i \tag{2}$$

$$= \mathbf{H}^i \boldsymbol{g} + \boldsymbol{v}^i + \boldsymbol{n}^i, \tag{3}$$

where the terms of the series expansion are arranged in the matrix $\mathbf{H}^i\left(r^i, \theta^i, \lambda^i\right)$, called shape matrix. The vector $\boldsymbol{v}^i$ describes the parts that are not parameterized by the underlying model, e.g. the external parts, closing the void between the parameterized



field and the measurement noise of the sensors, which is symbolized by the vector $\boldsymbol{n}^i$. Especially, the measurement noise is neither correlated with the parameterized part $\mathbf{H}^i \boldsymbol{g}$ nor with the non-parameterized part $\boldsymbol{v}^i$.

Summarizing the magnetic field measurements for all $N$ data points into a vector $\boldsymbol{B} = \left( \left( \boldsymbol{b}^1 \right)^T \dots \left( \boldsymbol{b}^N \right)^T \right)^T \in \mathbb{R}^{3N}$, the field can be written as

$$\boldsymbol{B} = \mathbf{H}\,\boldsymbol{g} + \boldsymbol{v} + \boldsymbol{n}, \tag{4}$$

where $\boldsymbol{n} = \left( \boldsymbol{n}^1 \dots \boldsymbol{n}^N \right)^T \in \mathbb{R}^{3N}$, $\boldsymbol{v} = \left( \boldsymbol{v}^1 \dots \boldsymbol{v}^N \right)^T \in \mathbb{R}^{3N}$ and $\mathbf{H} = \left[ \mathbf{H}^1 \dots \mathbf{H}^N \right]^T \in \mathbb{R}^{3N \times G}$. $G$ indicates the number of expansion coefficients.

Within Eq. (4) the magnetic field vector $\boldsymbol{B}$ and the shape matrix $\mathbf{H}$, given by the underlying model, are known. The coef-
ficient vector $\boldsymbol{g}$ is to be determined by data fitting. Since in most applications the number of known magnetic data points is much larger than the number of wanted expansion coefficients ($G \ll 3N$), $\mathbf{H}$ is a rectangular matrix in general. Furthermore, the non-parameterized parts of the field and the noise are unknown. Therefore, the direct inversion of Eq. (4) is impossible and $\boldsymbol{g}$ has to be estimated. In this case, Capon's method establishes a robust and useful tool to find the estimated solution for the expansion coefficients in Eq. (4). Since the method does not require the orthogonality of the basis functions, it has a wider
range of applications when decomposing the measured data into a set of superposed signals, especially when the number of data points is limited.

## 3    Derivation of Capon's method

The following derivation of Capon's method is based on the linear-algebraic formulation (Motschmann et al., 1996) which was formerly applied to the analysis of plasma waves in the terrestrial magnetosphere. Now we are focussing on the analysis of
planetary magnetic fields.

As illustrated in the previous section, the magnetic field $\boldsymbol{b}^i$ measured at data point $\boldsymbol{x}^i$ in the vicinity of Mercury and the wanted expansion coefficients $\boldsymbol{g}$ are related via

$$\boldsymbol{b}^i = \mathbf{H}^i \boldsymbol{g} + \boldsymbol{v}^i + \boldsymbol{n}^i \tag{5}$$

or in a compact form for all $N$ data points

$$\boldsymbol{B} = \mathbf{H}\,\boldsymbol{g} + \boldsymbol{v} + \boldsymbol{n}, \tag{6}$$

where the shape matrix $\mathbf{H}$ describes the underlying model.





For every data point $\boldsymbol{x}^i$ the noise vector $\boldsymbol{n}^i$ is assumed to be Gaussian with variance $\sigma_n$ and zero mean, so that $\langle \boldsymbol{n}^i \rangle = 0$ and $\langle \boldsymbol{n^i} \circ \boldsymbol{n^i} \rangle = \sigma_n^2 \mathbf{I}$, where $\mathbf{I}$ is the identity matrix. The angular brackets indicate averaging over an ensemble, e.g., different

samples, realizations or measurements. Therefore, $\langle \boldsymbol{b}^i \rangle = \mathbf{H}^i \, \boldsymbol{g} + \langle \boldsymbol{v}^i \rangle$ holds and equivalently

$$\langle \boldsymbol{B} \rangle = \mathbf{H} \, \boldsymbol{g} + \langle \boldsymbol{v} \rangle, \tag{7}$$

since the model $\mathbf{H}$ and the true coefficient vector $\boldsymbol{g}$ are not affected by the averaging.

Because $\mathbf{H}$ is not always a square matrix but is in general a rectangular matrix with different sizes between rows and

columns, the direct inversion of Eq. (7) is not guaranteed. Let us ignore for the moment the non-existence of $\mathbf{H}^{-1}$ and write down the equation

$$\boldsymbol{g} = \mathbf{H}^{-1} \left( \langle \boldsymbol{B} \rangle - \langle \boldsymbol{v} \rangle \right). \tag{8}$$

Despite its simplicity it is obviously incorrect. As $\mathbf{H}^{-1}$ does not exist, let us look for another matrix $\mathbf{w}$ called filter matrix, which follows the structure of this equation and fulfills in principle the resolution of Eq. (7) with respect to $\boldsymbol{g}$. Capon's method

is just the procedure to construct the filter matrix $\mathbf{w}$ and to calculate or better to say estimate $\boldsymbol{g}$. To do so, some helpful quantities are introduced. In order to distinguish between the true coefficient vector $\boldsymbol{g}$ and the estimated solution, in the following Capon's estimator will be symbolized by $\boldsymbol{g}_C$.

For the inversion of Eq. (4) the data covariance matrix $\mathbf{M}$ and the coefficient matrix $\mathbf{P}$ are introduced as follows:

$$\mathbf{M} = \langle \boldsymbol{B} \circ \boldsymbol{B} \rangle = \frac{1}{Q} \sum_{\alpha=1}^{Q} \boldsymbol{B}^\alpha \circ \boldsymbol{B}^\alpha \in \mathbb{R}^{3N \times 3N} \tag{9}$$

$$\mathbf{P} = \langle \boldsymbol{g}_C \circ \boldsymbol{g}_C \rangle \in \mathbb{R}^{G \times G} \tag{10}$$

Here $Q$ indicates the number of measurements and the circle '∘' symbolizes the outer product, which is defined by

$$\boldsymbol{x} \circ \boldsymbol{y} = \mathbf{x} \cdot \mathbf{y}^\dagger \in \mathbb{R}^{n \times m} \tag{11}$$

for any pair of vectors $\boldsymbol{x} \in \mathbb{R}^n$, $\boldsymbol{y} \in \mathbb{R}^m$. The dagger † indicates the Hermitian conjugate and the dot stands for the multipli-

cation of the matrices $\mathbf{x} \in \mathbb{R}^{n \times 1}$ and $\mathbf{y}^\dagger \in \mathbb{R}^{1 \times m}$. Therefore, the diagonal of the matrix $\mathbf{P}$ contains the quadratic averaged components of the wanted estimator. It is important to note here that the covariance matrix $\mathbf{M}$ must be statistically averaged, otherwise the matrix is singular with a vanishing determinant and the further analysis cannot be achieved.





If the non-invertibility of the matrix $\boldsymbol{B}^\alpha \circ \boldsymbol{B}^\alpha$ is neglected, for every measurement $\alpha = 1, \ldots, Q$ an estimator $\boldsymbol{g}_C^\alpha$ for the true

coefficient vector $\boldsymbol{g}$ can be determined, so that

$$\boldsymbol{B}^\alpha = \mathbf{H}\,\boldsymbol{g}_C^\alpha + \boldsymbol{v}^\alpha + \boldsymbol{n}^\alpha \tag{12}$$

is valid. Thereby, each estimator deviates from the true coefficient vector by an error vector $\varepsilon^\alpha = \boldsymbol{g} - \boldsymbol{g}_C^\alpha$. Note that because of

the non-invertibility of the matrix $\boldsymbol{B}^\alpha \circ \boldsymbol{B}^\alpha$ the single estimator $\boldsymbol{g}_C^\alpha$ cannot be calculated. Since the invertibility is solely given

by the averaging over $Q$ measurements, only the averaged estimator

$$\boldsymbol{g}_C = \frac{1}{Q} \sum_{\alpha=1}^{Q} \boldsymbol{g}_C^\alpha, \tag{13}$$

with its related error

$$\langle \varepsilon \rangle = \frac{1}{Q} \sum_{\alpha=1}^{Q} \varepsilon^\alpha \tag{14}$$

is available.

In contrast to the estimator, the true coefficient vector is a theoretical given vector, that is not affected by the averaging

($\boldsymbol{g} \equiv \langle \boldsymbol{g} \rangle$, Eq. (7)). This property directly links the estimator to the true coefficient vector, which can be rewritten as

$$\boldsymbol{g} = \boldsymbol{g}_C^\alpha + \varepsilon^\alpha. \tag{15}$$

Averaging over $Q$ measurements and using $\boldsymbol{g} \equiv \langle \boldsymbol{g} \rangle$ results in

$$\boldsymbol{g} = \boldsymbol{g}_C + \langle \varepsilon \rangle \tag{16}$$

and

$$\boldsymbol{g} \circ \boldsymbol{g} = \langle \boldsymbol{g}_C \circ \boldsymbol{g}_C \rangle + \langle \boldsymbol{g}_C \circ \varepsilon \rangle + \langle \varepsilon \circ \boldsymbol{g}_C \rangle + \langle \varepsilon \circ \varepsilon \rangle \tag{17}$$

analogously for the second order moments.

In the limit of vanishing errors $\langle \varepsilon \rangle \to 0$, $\langle \varepsilon \circ \varepsilon \rangle \to 0$ respectively, Capon's estimator converges to the true coefficient vector

$$\boldsymbol{g}_C \to \boldsymbol{g} \tag{18}$$

and therefore

$$\langle \boldsymbol{g}_C \circ \boldsymbol{g}_C \rangle \to \boldsymbol{g} \circ \boldsymbol{g}. \tag{19}$$





For the further evaluation of Eq. (4) the definition of the outer product (Eq. 11) is utilized. Matrix multiplication of Eq. (4) with its Hermitian adjoint and averaging yields

$$\langle \boldsymbol{B} \cdot \boldsymbol{B}^{\dagger} \rangle = \langle (\mathbf{H}\,\boldsymbol{g} + \boldsymbol{v} + \boldsymbol{n}) \cdot (\mathbf{H}\,\boldsymbol{g} + \boldsymbol{v} + \boldsymbol{n})^{\dagger} \rangle \tag{20}$$

and therefore

$$\langle \boldsymbol{B} \circ \boldsymbol{B} \rangle = \mathbf{H} \cdot \boldsymbol{g} \circ \boldsymbol{g} \cdot \mathbf{H}^{\dagger} + 2\,(\mathbf{H}\boldsymbol{g}) \circ \langle \boldsymbol{v} \rangle + \langle \boldsymbol{v} \circ \boldsymbol{v} \rangle + \langle \boldsymbol{n} \circ \boldsymbol{n} \rangle \qquad , \tag{21}$$

assuming, that $\boldsymbol{n}$ is Gaussian with variance $\sigma_n$ and zero mean ($\langle \boldsymbol{n} \rangle = 0$). By means of the limit $\langle \boldsymbol{\varepsilon} \rangle \to 0$ in Eq. (19) the unknown matrix $\boldsymbol{g} \circ \boldsymbol{g}$ and the true coefficient vector $\boldsymbol{g}$ can be approximated by Capon's estimator, resulting in

$$\langle \boldsymbol{B} \circ \boldsymbol{B} \rangle = \mathbf{H} \cdot \langle \boldsymbol{g}_C \circ \boldsymbol{g}_C \rangle \cdot \mathbf{H}^{\dagger} + 2\,(\mathbf{H}\boldsymbol{g}_C) \circ \langle \boldsymbol{v} \rangle + \langle \boldsymbol{v} \circ \boldsymbol{v} \rangle + \langle \boldsymbol{n} \circ \boldsymbol{n} \rangle. \tag{22}$$

Taking into account that $\langle \boldsymbol{n} \circ \boldsymbol{n} \rangle = \sigma_n^2 \mathbf{I}$ and using the above defined abbreviations, Eq. (22) can be rewritten as

$$\mathbf{M} = \mathbf{H}\mathbf{P}\mathbf{H}^{\dagger} + 2\,(\mathbf{H}\boldsymbol{g}_C) \circ \langle \boldsymbol{v} \rangle + \langle \boldsymbol{v} \circ \boldsymbol{v} \rangle + \sigma_n^2 \mathbf{I}. \tag{23}$$

Since Eq. (23) cannot be directly solved for $\mathbf{P}$, the goal is to find the best estimator $\boldsymbol{g}_C$ for $\boldsymbol{g}$, so that $\mathbf{P} = \langle \boldsymbol{g}_C \circ \boldsymbol{g}_C \rangle$ is obtained as an approximate solution of Eq. (23). Therefore, a filter matrix $\mathbf{w}$ is constructed, that separates the parameterized field from the noise by projecting the measured data onto the parameter space

$$\mathbf{w}^{\dagger} \langle \boldsymbol{B} \rangle = \boldsymbol{g}_C \tag{24}$$

and simultanously truncates the non-parameterized parts, i.e.

$$\mathbf{w}^{\dagger} \langle \boldsymbol{v} \rangle = 0. \tag{25}$$

Applying the filter matrix to the average of the non-parameterized parts of the field in Eq. (4)

$$0 = \mathbf{w}^{\dagger} \langle \boldsymbol{v} \rangle = \mathbf{w}^{\dagger} \left( \langle \boldsymbol{B} \rangle - \mathbf{H}\boldsymbol{g}_C \right) = \mathbf{w}^{\dagger} \langle \boldsymbol{B} \rangle - \mathbf{w}^{\dagger} \mathbf{H}\boldsymbol{g}_C = \boldsymbol{g}_C - \mathbf{w}^{\dagger} \mathbf{H}\boldsymbol{g}_C, \tag{26}$$

where the true coefficient vector has been replaced by Capon's estimator (Eq. 16) and taking into account that $\langle \boldsymbol{n} \rangle = 0$ leads to the distortionless constraint

$$\mathbf{w}^{\dagger} \mathbf{H} = \mathbf{I}. \tag{27}$$

This equation is one of the important constraints for the construction of the wanted filter matrix $\mathbf{w}$ but it is not enough. Let us look for another criterion. The basic idea is that in Eq. (4) there may be contributions $\langle \boldsymbol{v} \rangle$ in the data $\boldsymbol{B}$ which are not caused by the internal magnetic field and thus, these contributions are not modeled by $\mathbf{H}\boldsymbol{g}$. Although the filter matrix $\mathbf{w}$ is already designed to truncate these parts, i.e. $\mathbf{w}^{\dagger} \langle \boldsymbol{v} \rangle = 0$, their contributions to the data are unknown. This yields the following procedure.





Conferring to Eq. (24) the average output power, which is defined as the sum of the quadratic averaged components of the
estimator, can be rewritten as (Pillai , 1989)

$$\operatorname{tr} \mathbf{P} = \operatorname{tr} \langle \boldsymbol{g}_C \circ \boldsymbol{g}_C \rangle = \operatorname{tr} \left( \mathbf{w}^\dagger \langle \boldsymbol{B} \circ \boldsymbol{B} \rangle \, \mathbf{w} \right), \tag{28}$$

where $\operatorname{tr} \langle \boldsymbol{g}_C \circ \boldsymbol{g}_C \rangle$ is the trace of the matrix $\langle \boldsymbol{g}_C \circ \boldsymbol{g}_C \rangle$.

Using Eq. (28), the coefficient matrix $\mathbf{P}$ can be expressed by the weight $\mathbf{w}$ and the data covariance $\mathbf{M}$ as

$$\mathbf{P} = \langle \boldsymbol{g}_C \circ \boldsymbol{g}_C \rangle = \mathbf{w}^\dagger \langle \boldsymbol{B} \circ \boldsymbol{B} \rangle \, \mathbf{w} = \mathbf{w}^\dagger \mathbf{M} \mathbf{w}. \tag{29}$$

Since the amount of the data's noise and the amount of the non-parameterized parts are unknown, we assume here, that a large
part of the data are influenced by the noise and the non-parameterized parts, that have to be truncated by the matrix $\mathbf{w}$ and the
underlying model keeps the minimal contribution to the data. This minimal contribution has to be extracted.

Therefore, the output power $P = \operatorname{tr} \mathbf{P}$ has to be minimized with respect to $\mathbf{w}^\dagger$, subject to the distortionless constraint $\mathbf{w}^\dagger \mathbf{H} = \mathbf{I}$ or equivalently $\mathbf{H}^\dagger \mathbf{w} = \mathbf{I}$ . Using the Lagrange multiplier method this minimization problem can be formulated as

$$\text{minimize} \quad \operatorname{tr} \left[ \mathbf{w}^\dagger \mathbf{M} \mathbf{w} + \boldsymbol{\Lambda} \left( \mathbf{I} - \mathbf{H}^\dagger \mathbf{w} \right) \right], \tag{30}$$

where $\boldsymbol{\Lambda}$ are the related Lagrange multipliers and the minimum is taken with respect to $\mathbf{w}$. Since the components $w_{ij}$ and $w_{ij}^\dagger$ of the matrix $\mathbf{w}$ and $\mathbf{w}^\dagger$ respectively, are independent of each other, Eq. (30) can be expanded as

$$\text{minimize} \quad P = \operatorname{tr} \left[ \mathbf{w}^\dagger \mathbf{M} \mathbf{w} + \boldsymbol{\Lambda} \left( \mathbf{I} - \mathbf{H}^\dagger \mathbf{w} \right) + \left( \mathbf{I} - \mathbf{w}^\dagger \mathbf{H} \right) \boldsymbol{\Gamma} \right] \tag{31}$$

or equivalently

$$\text{minimize} \quad P = w_{ij}^\dagger M_{jk} w_{ki} + \Lambda_{ii} - \Lambda_{ij} H_{jk}^\dagger w_{ki} + \Gamma_{ii} - w_{ij}^\dagger H_{jk} \Gamma_{ki} \tag{32}$$

with related additional Lagrange multipliers $\boldsymbol{\Gamma}$. Taking the derivatives with respect to $w_{ki}$ and $w_{ij}^\dagger$ results in

$$0 = \partial_{w_{ki}} P = w_{ij}^\dagger M_{jk} - \Lambda_{ij} H_{jk}^\dagger \tag{33}$$

yielding

$$\mathbf{w}^\dagger \mathbf{M} = \boldsymbol{\Lambda} \mathbf{H}^\dagger \tag{34}$$

and

$$0 = \partial_{w_{ij}^\dagger} P = M_{jk} w_{ki} - H_{jk} \Gamma_{ki} \tag{35}$$





resulting in

$$\mathbf{M}\mathbf{w} = \mathbf{H}\boldsymbol{\Gamma} \quad . \tag{36}$$

Multiplication of Eq. (34) with $\mathbf{w}$ from the right and multiplication of Eq. (36) with $\mathbf{w}^{\dagger}$ from the left considering the distortionless constraint delivers

$$\mathbf{P} = \mathbf{w}^{\dagger}\mathbf{M}\mathbf{w} = \boldsymbol{\Gamma} = \boldsymbol{\Lambda} \tag{37}$$

and therefore

$$P = \operatorname{tr}\mathbf{P} = \operatorname{tr}\boldsymbol{\Gamma} = \operatorname{tr}\boldsymbol{\Lambda}. \tag{38}$$

Because $P = \operatorname{tr}\mathbf{P}$ is a convex function, $\boldsymbol{\Lambda}$ and $\boldsymbol{\Gamma}$ are realizing the minimal output power.

Due to the ensemble averaging the matrix $\mathbf{M}$ is invertible and $\mathbf{M}^{-1}$ exists. Multiplying Eq. (36) with $\mathbf{H}^{\dagger}\mathbf{M}^{-1}$ and again considering the distortionless constraint yields

$$\mathbf{P} = \boldsymbol{\Lambda} = \boldsymbol{\Gamma} = \left[\mathbf{H}^{\dagger}\mathbf{M}^{-1}\mathbf{H}\right]^{-1}. \tag{39}$$

By means of Eq. (34) the filter matrix results in

$$\mathbf{w}^{\dagger} = \mathbf{P}\mathbf{H}^{\dagger}\mathbf{M}^{-1} \tag{40}$$

and therefore Capon's estimator is given by

$$\boldsymbol{g}_C = \left[\mathbf{H}^{\dagger}\mathbf{M}^{-1}\mathbf{H}\right]^{-1}\mathbf{H}^{\dagger}\mathbf{M}^{-1}\langle\boldsymbol{B}\rangle. \tag{41}$$

Regarding the expensive derivation, this compact formular for Capon's estimator is surprising. The same expression can be derived by treating Capon's method as a special case of the maximum likelihood estimator (Narita, 2019).

Since Capon's method has formerly been applied to the analysis of waves, the existing derivations treat the non-parameterized parts of the field as Gaussian noise. Considering the analysis of planetary magnetic fields this assumption is indefensible. For 205 example, the external parts of the field are systematic noise and cannot be modeled by a Gaussian distribution. Therefore, the above presented derivation generalizes the previous derivations of Capon's method.

## 4 Diagonal Loading

The filter matrix $\mathbf{w}$ is the key parameter to distinguish between the parameterized and the non-parameterized parts of the field. Conferring to Eq. (23) the ratio of these parts defines the input signal-noise-ratio

$$SNR_i = \frac{\operatorname{tr}\left(\mathbf{H}\mathbf{P}\mathbf{H}^{\dagger}\right)}{\operatorname{tr}\left(2\left(\mathbf{H}\boldsymbol{g}_C\right)\circ\langle\boldsymbol{v}\rangle + \langle\boldsymbol{v}\circ\boldsymbol{v}\rangle + \sigma_n^2\mathbf{I}\right)}. \tag{42}$$





The filter matrix is applied to the disturbed data for estimating the output power, that is related to Capon's estimator. Thus, the output signal-noise-ratio can be expressed as (Haykin, 2014; Van Trees, 2002)

$$SNR_o = \frac{\mathrm{tr}\left(\mathbf{w}^\dagger \mathbf{H} \mathbf{P} \mathbf{H}^\dagger \mathbf{w}\right)}{\mathrm{tr}\left(2\mathbf{w}^\dagger \left(\mathbf{H}\boldsymbol{g}_C\right) \circ \langle\boldsymbol{v}\rangle \mathbf{w} + \mathbf{w}^\dagger \langle\boldsymbol{v} \circ \boldsymbol{v}\rangle \mathbf{w}\right) + \sigma_n^2 \mathrm{tr}\left(\mathbf{w}^\dagger\mathbf{w}\right)} = \frac{\mathrm{tr}\left(\mathbf{P}\right)}{\sigma_n^2 \mathrm{tr}\left(\mathbf{w}^\dagger\mathbf{w}\right)}, \tag{43}$$

since the filter fulfills the distortionless constraint $\mathbf{w}^\dagger\mathbf{H} = \mathbf{I}$ and truncates the non-parameterized parts of the field, i.e. $\mathbf{w}^\dagger\langle\boldsymbol{v}\rangle =$

0. The ratio of the output and the input signal-noise-ratio is the so-called array gain (Van Trees, 2002)

$$\frac{SNR_o}{SNR_i} = \frac{1}{\mathrm{tr}\left(\mathbf{w}^\dagger\mathbf{w}\right)} \frac{\mathrm{tr}\left(\mathbf{P}\right)}{\sigma_n^2 \mathrm{tr}\left(\mathbf{H}\mathbf{P}\mathbf{H}^\dagger\right)} \mathrm{tr}\left(2\left(\mathbf{H}\boldsymbol{g}_C\right) \circ \langle\boldsymbol{v}\rangle + \langle\boldsymbol{v} \circ \boldsymbol{v}\rangle + \sigma_n^2\mathbf{I}\right) \sim \frac{1}{\mathrm{tr}\left(\mathbf{w}^\dagger\mathbf{w}\right)}, \tag{44}$$

which is controlled by $\mathrm{tr}\left(\mathbf{w}^\dagger\mathbf{w}\right)$.

The input signal-noise-ratio is given by the data and the underlying model and therefore, $SNR_i = \mathrm{const.}$ in Eq. (44). In

contrast to the input signal-noise-ratio, the output signal-noise-ratio depends on the filtering. When $\mathrm{tr}\left(\mathbf{w}^\dagger\mathbf{w}\right)$ is large, the output signal-noise-ratio can decrease ($SNR_o \to 0$), resulting in signal elimination and thus, the performance of Capon's estimator degrades. To prevent the signal elimination and to improve the robustness of Capon's estimator it is desirable to restrict $\mathrm{tr}\left(\mathbf{w}^\dagger\mathbf{w}\right)$ with an upper boundary $T_0 = \mathrm{const.}$ (Van Trees, 2002), which can be expressed by the additional quadratic constraint

$$\mathrm{tr}\left(\mathbf{w}^\dagger\mathbf{w}\right) = T_0. \tag{45}$$

For reasons of mathematical aesthetics, the constant $T_0$ is expressed as the trace of a matrix $\mathbf{T}$. For example, one can choose

$$\mathbf{T} = \frac{T_0}{G}\mathbf{I} \in \mathbb{R}^{G \times G}, \tag{46}$$

where $G$ again indicates the number of wanted expansion coefficients and $\mathbf{I} \in \mathbb{R}^{G \times G}$ is the identity matrix, so that

$$\mathrm{tr}\left(\mathbf{T}\right) = \frac{T_0}{G}\mathrm{tr}\left(\mathbf{I}\right) = T_0 \tag{47}$$

holds. Thus, Eq. (45) can be rewritten as

$$\mathrm{tr}\left(\mathbf{w}^\dagger\mathbf{w} - \mathbf{T}\right) = 0. \tag{48}$$

It should be noted that the matrix $\mathbf{T}$ can be chosen arbitrarily as long as it is independent of $\mathbf{w}$ and $\mathbf{w}^\dagger$.

Conferring to the previous section and considering the additional quadratic constraint, the filter matrix can be calculated by

solving

$$\mathrm{minimize} \quad \mathrm{tr}\left[\mathbf{w}^\dagger\mathbf{M}\mathbf{w} + \sigma_d^2\left(\mathbf{w}^\dagger\mathbf{w} - \mathbf{T}\right) + \boldsymbol{\Lambda}\left(\mathbf{I} - \mathbf{H}^\dagger\mathbf{w}\right) + \left(\mathbf{I} - \mathbf{w}^\dagger\mathbf{H}\right)\boldsymbol{\Gamma}\right] \tag{49}$$

with respect to $\mathbf{w}$, where $\sigma_d^2$ is the related additional Lagrange multiplier. Carrying out the same procedure as described in section 3, the constrained minimizer is given by

$$\mathbf{w} = \left(\mathbf{M} + \sigma_d^2\mathbf{I}\right)^{-1}\mathbf{H}\left[\mathbf{H}^\dagger\left(\mathbf{M} + \sigma_d^2\mathbf{I}\right)^{-1}\mathbf{H}\right]^{-1}. \tag{50}$$





The comparison of Eq. (40) with Eq. (50) shows that the quadratic constraint results in the addition of the constant value $\sigma_d^2$ to the diagonal of the data covariance matrix $\mathbf{M}$, which is known as diagonal loading (Van Trees, 2002). Consequently, the filter matrix is designed for a higher Gaussian background noise than is actually present (Van Trees, 2002).

In Figure 1 $\mathrm{tr}\left(\mathbf{w}^\dagger\mathbf{w}\right)$ is displayed with respect to $\sigma = \sqrt{\sigma_d^2 + \sigma_n^2}$. For $\sigma_d \to 0$, $\mathrm{tr}\left(\mathbf{w}^\dagger\mathbf{w}\right)$ is large resulting in a small output
signal-noise ratio, which can cause signal elimination. For increasing values of $\sigma_d$, $\mathrm{tr}\left(\mathbf{w}^\dagger\mathbf{w}\right)$ decreases and for $\sigma_d \to \infty$ Capon's filter converges to the least square fit filter

$$\mathbf{w}^\dagger \stackrel{\sigma_d \to \infty}{\longrightarrow} \left[\mathbf{H}^\dagger\mathbf{H}\right]^{-1}\mathbf{H}^\dagger \tag{51}$$

or equivalently

$$\mathbf{w}^\dagger\mathbf{w} \stackrel{\sigma_d \to \infty}{\longrightarrow} \left[\mathbf{H}^\dagger\mathbf{H}\right]^{-1}, \tag{52}$$

that treats all data equally.

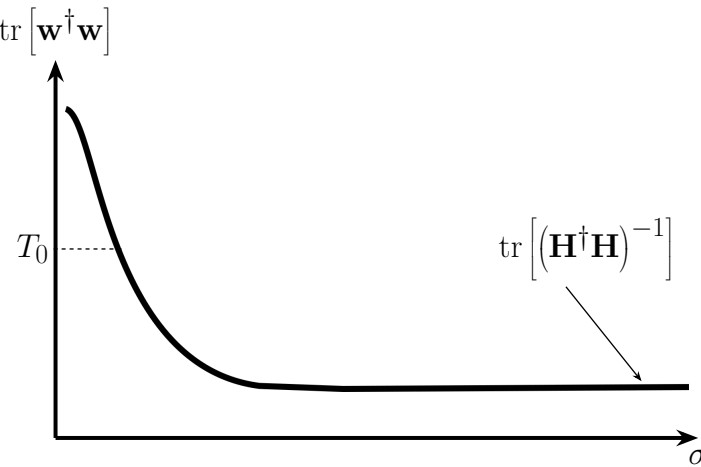

**Figure 1.** Sketch of the trace of the filter matrix with respect to $\sigma = \sqrt{\sigma_d^2 + \sigma_n^2}$. For increasing values of the diagonal loading parameter $\sigma_d$ the output signal-noise ratio increases and Capon's filter converges to the least square fit filter, if the diagonal loading parameter is large.

Since $\mathrm{tr}\left(\mathbf{w}^\dagger\mathbf{w}\right)$ is a monotonically decreasing function, $\sigma_d \to \infty$ might be the best choice for the diagonal loading parameter. To check this expectation we have to take a look at the output power $P_d$ for the synthetic increased noise, which can be estimated by replacing $\mathbf{M} \to \mathbf{M} + \sigma_d^2\mathbf{I}$ in Eq. (39), resulting in (Pajovic, 2019; Richmond et al., 2005)

$$P_d = \mathrm{tr}\left[\mathbf{w}^\dagger\left(\mathbf{M}+\sigma_d^2\mathbf{I}\right)\mathbf{w}\right] = \mathrm{tr}\left[\mathbf{H}^\dagger\left(\mathbf{M}+\sigma_d^2\mathbf{I}\right)^{-1}\mathbf{H}\right]^{-1} \stackrel{\sigma_d \to \infty}{\longrightarrow} \sigma_d^2\,\mathrm{tr}\left[\mathbf{H}^\dagger\mathbf{H}\right]^{-1}. \tag{53}$$





Thus, $P_d$ is an increasing function of $\sigma_d$. Since the output power has to be minimized one would expect that $\sigma_d = 0$ is the best choice, which is in contradiction to the argumentation above. Therefore, the maximization of the array gain is not equivalent to the minimization of the output power (Van Trees, 2002). Since $\sigma_d$ cannot be directly calculated within the minimization procedure (Eq. 49), it is not clear how to choose the optimal diagonal loading parameter $\sigma_{opt.}$, that lies somewhere between those extrema.

In the literature there exist several methods for determining the optimal diagonal loading parameter (Pajovic, 2019). In contrast to the analysis of waves, we favor measurements that are stationary up to the Gaussian noise for the analysis of planetary magnetic fields. Comparing the measurement times with planetary geology time scales this assumption is surely valid for the internal magnetic field. For the external parts of the field this can be realized by choosing data sets of preferred situations, e.g. calm solar wind conditions. By virtue of the stationarity, the data covariance matrix $\mathbf{M} = \langle \boldsymbol{B} \rangle \circ \langle \boldsymbol{B} \rangle + \sigma_n^2 \mathbf{I}$ contains only one non-trivial eigenvalue $\lambda_1 = \left| \langle \boldsymbol{B} \rangle \right|^2 + \sigma_n^2$ and $\lambda_i = \sigma_n^2$, for $i = 2, \ldots, 3N$ elsewhere. Therefore, estimators for the diagonal loading parameter, that are related to eigenvalues corresponding to interference and noise (Carlson, 1988) or estimators taking into account the standard deviation of the diagonal elements of the data covariance matrix (Ma and Goh, 2003) cannot be applied. When simulated data are analyzed the deviation between Capon's estimator and the true coefficient vector, implemented in the simulation, is a useful metric for estimating the optimal diagonal loading parameter (Pajovic, 2019; Toepfer et al., 2020). But when the method is applied to real spacecraft data no true coefficient vector is available and therefore, another estimation method for the diagonal loading parameter, that solely depends on the data and the underlying model is required.

The additional quadratic constraint (Eq. 45), resulting in diagonal loading, bounds the trace of the filter matrix, that is the solution of the minimization procedure (Eq. 49). To prevent signal elimination, $\mathrm{tr}\left(\mathbf{w}^\dagger \mathbf{w}\right)$ and $P_d$ have to be minimized simultanously. Since $\mathrm{tr}\left(\mathbf{w}^\dagger \mathbf{w}\right)$ is decreasing for higher values of $\sigma_d$ (cf. Figure 1), $P_d$ is an increasing function and thus, they act as competitors. At the optimal value $\sigma_{opt.}$ the two competitors compromise. Therefore, the diagonal loading of the data covariance matrix is equivalent to the Tikhonov regularization for ill-posed problems and the optimal diagonal loading parameter can be estimated analogously by the method of the L-curve (Hiemstra et al. (2002)). The L-curve arises by plotting $\lg\left[\mathrm{tr}\left(\mathbf{w}^\dagger \mathbf{w}\right)\right]$ versus $\lg\left[\mathrm{tr}\left(\mathbf{w}^\dagger \left(\mathbf{M} + \sigma_d^2 \mathbf{I}\right)\mathbf{w}\right)\right]$ for different values of $\sigma_d$ and is displayed in Figure 2. The optimal value $\sigma_{opt.}$ is located in the vicinity of the L-curve's knee, which is defined by the maximum curvature of the L-curve (Hiemstra et al., 2002).

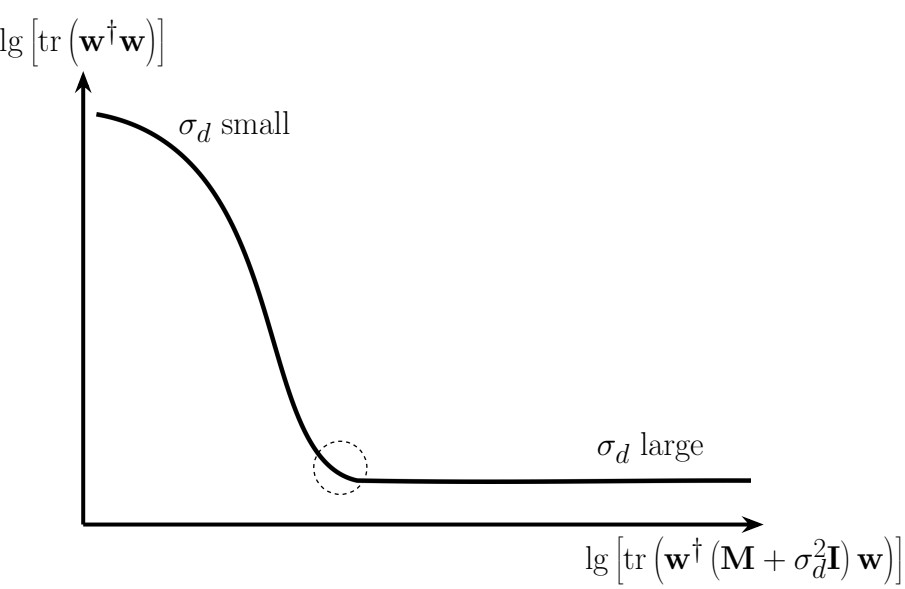

**Figure 2.** Sketch of the L-curve for estimating the optimal diagonal loading parameter $\sigma_d$. The optimal value is located in the vicinity of the L-curves knee (dashed circle).





## 5 Practical application of Capon's method

When Capon's method is applied to the reconstruction of Mercury's internal magnetic field (Toepfer et al., 2020), several computationally burdensome matrix inversions are necessary to calculate the estimator

$$\boldsymbol{g}_C = \left[\mathbf{H}^\dagger \mathbf{M}^{-1} \mathbf{H}\right]^{-1} \mathbf{H}^\dagger \mathbf{M}^{-1} \langle \boldsymbol{B} \rangle. \tag{54}$$

Thus, for the practical application of Capon's method it is useful to rewrite the method in terms of the least square fit method.

Capon's estimator inherits the matrix operation structure of the least square fit estimator, which is given by (Haykin, 2014)

$$\boldsymbol{g}_L = \left[\mathbf{H}^\dagger \mathbf{H}\right]^{-1} \mathbf{H}^\dagger \langle \boldsymbol{B} \rangle. \tag{55}$$

Substituting $\mathbf{H} \to \mathbf{M}^{-1/2}\mathbf{H}$ and $\langle \boldsymbol{B} \rangle \to \mathbf{M}^{-1/2}\langle \boldsymbol{B} \rangle$ in Eq. (55), the least square fit estimator may be converted into Capon's estimator.

The least square fit method minimizes the deviaton between the disturbed measurements $\boldsymbol{B}$ and the underlying model $\mathbf{H}\boldsymbol{g}$, measured in the Euclidian norm $||.||_2$, so that

$$\boldsymbol{g}_L = \arg\min_{\boldsymbol{g}} \left|\left|\mathbf{H}\boldsymbol{g} - \boldsymbol{B}\right|\right|_2^2. \tag{56}$$

    Conferring to the above mentioned substitutions, Capon's method can be interpreted as measuring the deviation in Eq. (56) in a different norm

$$\left|\left|\mathbf{H}\boldsymbol{g} - \boldsymbol{B}\right|\right|_{\mathbf{M}^{-1}}^2 = \left|\left|\mathbf{M}^{-1/2}\left(\mathbf{H}\boldsymbol{g} - \boldsymbol{B}\right)\right|\right|_2^2 \quad, \tag{57}$$

so that Capon's estimator is given by

$$\boldsymbol{g}_C = \arg\min_{\boldsymbol{g}} \left|\left|\mathbf{H}\boldsymbol{g} - \boldsymbol{B}\right|\right|_{\mathbf{M}^{-1}}^2. \tag{58}$$

    Thus, Capon's method can be regarded as a special case of the least square fit method or more precisely of the weighted least square fit method, where the data and the model are measured and weighted with the inverse data covariance matrix.
This property is useful for the practical application of Capon's method. In contrast to the computationally burdensome matrix inversions in Eq. (41), the usage of minimization algorithms, such as gradient descent or conjugate gradient method, for solving Eq. (58) is more stable and computationally inexpensive.





To illustrate the above presented mathematical foundations, Capon's method is applied to simulated magnetic field data in order to reconstruct Mercury's internal magnetic field. As a proof of concept, the underlying model is restricted to the internal

dipole and quadrupole contributions to the magnetic field as discussed in section 2.

For the reconstruction of Mercury's internal dipole and quadrupole field the internal Gauss coefficients $g_1^0 = -190\,\mathrm{nT}$ and $g_2^0 = -78\,\mathrm{nT}$ (Anderson et al., 2012; Wardinski et al., 2019), defining the non-vanishing components of the true coefficient vector $\boldsymbol{g}$ are implemented in the hybrid code AIKEF (Müller et al., 2011) and the magnetic field data resulting from the

plasma interaction of Mercury with the solar wind are simulated. The data are evaluated along selected parts of the prospective trajectories of the BepiColombo mission on the night side of Mercury within a distance of $0.2\,R_\mathrm{M}$ up to $0.4\,R_\mathrm{M}$ from Mercury's surface. Since simulated data are analyzed, the deviation between the true coefficient vector $\boldsymbol{g}$ and Capon's estimator $\boldsymbol{g}_C$ can be used as a metric to verify the estimation of the optimal diagonal loading parameter by making use of the L-curve technique. The optimal diagonal loading parameter results in $\sigma_{opt.} \approx 800\,\mathrm{nT}$ which corresponds with the vicinity of the L-curve's knee.

The reconstructed Gauss coefficients for the internal dipole and quadrupole field are presented in Table 1.

**Table 1.** Implemented and reconstructed Gauss coefficients for the internal dipole and quadrupole field.

| Gauss coefficient | input in nT | output Capon in nT |
|:---:|:---:|:---:|
| $g_1^0$ | -190.0 | -189.2 |
| $g_1^1$ | 0.0 | 1.9 |
| $h_1^1$ | 0.0 | 0.2 |
| $g_2^0$ | -78.0 | -68.4 |
| $g_2^1$ | 0.0 | 26.1 |
| $h_2^1$ | 0.0 | 11.4 |
| $g_2^2$ | 0.0 | -2.4 |
| $h_2^2$ | 0.0 | 0.0 |

The deviation $\left| \boldsymbol{g}_C - \boldsymbol{g} \right|$ between Capon's estimator and the true coefficient vector results in $30.2\,\mathrm{nT}$ or $14.7\,\%$, respectively, for the optimal diagonal loading parameter. When the magnetic field data are evaluated at an ensemble of data points with a distance of $0.2\,R_\mathrm{M}$ from Mercury's surface this deviation is of the same order (Toepfer et al., 2020). Extending the underlying model by a parameterization of the external parts of the magnetic field improves Capon's estimator especially when the

magnetic field data are evaluated in some distance above Mercury's surface.





# 6   Conclusions

Capon's method is a robust and useful tool for various kinds of ill-posed inverse problems, such as Mercury's planetary magnetic field analysis. The derivation of the method can be regarded from different mathematical perspectives. Here we revisited the linear-algebraic matrix formulation of the method and extended the derivation for Mercury's magnetic field analysis.

Capon's method becomes even more robust by incorporating the diagonal loading technique. Thereby, the construction of a filter matrix is vital to the derivation of Capon's estimator.

Especially the trace of the filter matrix determines the array gain, which is defined as the ratio of the output and the input signal-noise-ratio. If the trace is large, the output signal-noise-ratio can decrease resulting in signal elimination and thus, the

performance of Capon's estimator degrades. Bounding the trace of the filter matrix results in diagonal loading of the data covariance matrix, which improves the robustness of the method. The main problem of the diagonal loading technique is that in general it is not clear how to choose the optimal diagonal loading parameter. Since for the analysis of planetary magnetic fields we prefer measurements that are stationary up to the Gaussian noise, estimators for the diagonal loading parameter, that are related to eigenvalues of the data covariance matrix corresponding to interference and noise cannot be applied. Making use

of the L-curve's technique enables a robust procedure for estimating the optimal diagonal loading parameter.

For the calculation of Capon's estimator several computationally burdensome matrix inversions are necessary. Interpretation of Capon's method as a special case of the least square fit method enables the usage of numerically more stable and less burden minimization algorithms, e.g. gradient descent or conjugate gradient method, for calculating Capon's estimator.


It should be noted that the parameterization of Mercury's internal magnetic field via the Gauss representation, as mentioned in section 2, is only one of several possibilities for modeling the magnetic field in the vicinity of Mercury. The underlying model can be extended by other parameterizations, for example the Mie representation (toroidal-poloidal decomposition) or magnetospheric models and Capon's method can be applied to estimate the related model coefficients. Besides the analysis

of Mercury's internal magnetic field, the extention of the model also enables the reconstruction of current systems flowing in the magnetosphere. Concerning the BepiColombo mission this work establishes a mathematical basis for the application of Capon's method to analyze Mercury's internal magnetic field in a robust and manageable way.

*Data availability.*   The raw data supporting the conclusions of this article will be made available by the authors, without undue reservation.

*Author contributions.*   All authors contributed conception and design of the study; ST, YN and UM wrote the first draft of the manuscript;

All authors contributed to manuscript revision, read and approved the submitted version.





*Funding:* We acknowledge support by the German Research Foundation and the Open Access Publication Funds of the Technische Universität Braunschweig.

The work by Y. Narita is supported by the Austrian Space Applications Programme at the Austrian Research Promotion Agency under contract 865967.

D. Heyner was supported by the German Ministerium für Wirtschaft und Energie and the German Zentrum für Luft- und Raumfahrt under contract 50 QW1501.

*Competing interests.* The authors declare that they have no conflict of interest.

*Acknowledgements.* The authors are grateful for stimulating discussions and helpful suggestions by Karl-Heinz Glassmeier and Alexander Schwenke.



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
