# Peer review of "Mathematical foundation of Capon's method for planetary magnetic field analysis"

_Geoscientific Instrumentation, Methods and Data Systems, 2020_

## Referee Comment (RC1) · Anonymous Referee #1 · 7 Sep 2020

The paper *Mathematical foundation of Capon's method for planetary magnetic field analysis* provides the underlying formalism for applying Capon's method to planetary magnetic fields and illustrates it with simulated data relevant for BepiColombo mission. While this is a valuable contribution to the field, a number of points need to be addressed before publication.

1. *Extension of Capon's method to planetary magnetic fields.* As indicated in the first and last paras of Section 3, L73–75 and L202–206, the paper generalizes the Capon method, previously used for the analysis of wave data. This is a major result that could be emphasized better, perhaps in a separate discussion section. This section could include a closer analysis of the case in the paper as compared to the wave case, by referring to Motschmann et al. (1996).

The discussion section could also detail the key principle(s) underlying the method, like maximum likelihood / minimum variance, along the line of Narita (2019). The divergence free feature of the magnetic field could be discussed on top, as in Motschmann et al. (1996).

Related:
- L166-168: I am not sure I understand the text here, even though I could essentially follow Eqs. 30 to 41. Eventually, the underlying model makes the main contribution to the data (e.g., in the test application of Section 5). I guess this 'minimal contribution' has rather the meaning of Motschmann et al. (1996), where the filter w absorbs all the energy not associated with k (here not associated with the parametrized field) and leaves the part related to k undistorted (here the part related to the parametrized field). Same issue at L256.
- L200-201: Is this expression derived by Narita (2019)? Or could be derived by further processing of the maximum likelihood estimator? (e.g., in the suggested discussion section?)

2. *Illustration & Validation.* In view of upcoming BepiColombo data, the authors chose to illustrate the method with simulated observations of Mercury magnetic field. While this is certainly helpful to prepare BepiColombo, I wonder if it is also the best test bed for the method. Earth magnetic field is known much better, at various altitudes – such that the weight of the external field and its influence on the results could be analyzed too. Including an example at the Earth, or at least a brief discussion of this validation possibility, would be more than welcomed.

Regarding the test exercise of Section 5, Table 1 shows that the largest errors are associated with $g_2^1$ and, to some extent, with $h_2^1$. Is this by chance, or related to some systematics?

3. *Technicalities.* Considering the target audience of the journal, different to a good extent from the signal processing community, the mathematical language of the paper may prevent the optimal transmission of the message. Additional explanations may help, inserted in the text or collected in an Appendix – when detailing the math would perturb the flow too much:
- L69-71: Please clarify this sentence, possibly including an example.
- L93-97: The intuitive introduction of the filter matrix w via Eq. 8 s a bit confusing, since eventually the non-parametrized (external) part of B does not show up in the $g_C$ formula, Eq. 41.
- Eqs. 9 and 10 fall pretty much out of the blue. The use of M and P becomes clear later, but some clarification would be good already at this point.
- L106-110: Please detail why the determinant vanishes (even though it may look straight), how does statistical average prevent this, how is statistical average achieved.
- L127: Please indicate also the second order moments.
- Eq. 21: Please explain why 2<Hg> o <v> and not <Hg> o <v> + <v> o <Hg> (given that, in general, the external product does not commute).
- Eq. 27 and L154: Please explain why this is not enough to uniquely determine w.
- L155-158: Feels confuse. As long as the filter matrix truncates the non-parametrized part, it is not clear why its contribution to the data matters, neither how 'this yields the following procedure'.
- L191: Why is tr P a convex function?

- Eqs. 42 and 43: Please detail what is meant by 'input' and 'output'. Regarding Eq. 43, is there an equation analogous to Eq. 23, to clarify the meaning of 'signal' and 'noise' also for output?
- Eq. 44: Please explain why this ratio is dominated by 1/trace.
- L266-267: Please provide a brief demonstration.
- L267-269: Please explain briefly what is this about.
- L278: How is the 'compromise' quantified?
- L278-282: This is quite opaque for those not familiar with signal processing and in particular with these techniques.

4. *Others*
- L17-18: What is non-ideal orbits?
- L18: simulated Mercury magnetic field data
- L54: 'closing the void' => 'covering the range' ?
- Eq. 54 is identical to Eq. 41.
- L325: in => at
- L352: In principle, one could analyze also the external field, if some model is adopted.

---

## Author Comment (AC1) · 1 Oct 2020

*The paper Mathematical foundation of Capon's method for planetary magnetic field analysis provides the underlying formalism for applying Capon's method to planetary magnetic fields and illustrates it with simulated data relevant for BepiColombo mission. While this is a valuable contribution to the field, a number of points need to be addressed before publication.*

*1. Extension of Capon's method to planetary magnetic fields. As indicated in the first and last paras of Section 3, L73–75 and L202–206, the paper generalizes the Capon method, previously used for the analysis of wave data. This is a major result that could be emphasized better, perhaps in a separate discussion section. This section could include a closer analysis of the case in the paper as compared to the wave case, by referring to Motschmann et al. (1996).*
*The discussion section could also detail the key principle(s) underlying the method, like maximum likelihood / minimum variance, along the line of Narita (2019). The divergence free feature of the magnetic field could be discussed on top, as in Motschmann et al. (1996).*

**Reply:** Agreed. We will add a section dedicated to the connection of the Capon method to other inversion methods (e.g., maximum likelihood, least square fit). We will highlight the difference from the wave analysis method (Motschmann et al., 1996) in section 6. It is also a good idea to add a discussion about the divergence-free nature of the magnetic field in the introduction.

*Related:*
*• L166-168: I am not sure I understand the text here, even though I could essentially follow Eqs. 30 to 41. Eventually, the underlying model makes the main contribution to the data (e.g., in the test application of Section 5). I guess this 'minimal contribution' has rather the meaning of Motschmann et al. (1996), where the filter w absorbs all the energy not associated with k (here not associated with the parametrized field) and leaves the part related to k undistorted (here the part related to the parametrized field). Same issue at L256.*

**Reply:** Surely it is possible that the underlying model makes the main contribution to the data. But the distribution of the parameterized and the non-parameterized parts is unknown. Therefore, we stay conservative and assume safely that a large part of data is influenced by the noise and non-parametrized signals. (Explanation will be added on page 7)

*• L200-201: Is this expression derived by Narita (2019)? Or could be derived by further processing of the maximum likelihood estimator? (e.g., in the suggested discussion section?)*

**Reply:** Yes, Narita (2019) derived the expression for Capon's estimator by regarding the likelihood function as nearly Gaussian (particularly around the peak of the likelihood function).

*2. Illustration & Validation. In view of upcoming BepiColombo data, the authors chose to illustrate the method with simulated observations of Mercury magnetic field. While this is certainly helpful to prepare BepiColombo, I wonder if it is also the best test bed for the method. Earth magnetic field is known much better, at various altitudes – such that the weight of the external field and its influence on the results could be analyzed too. Including an example at the Earth, or at least a brief discussion of this validation possibility, would be more than welcomed.*

**Reply:** The application of Capon's method to the analysis of Mercury's magnetic field will been emphasized as an example. Surely, the test against the Earth' magnetic field would be a good alternative. But such an analysis would have the size of a separate paper and it could not be pressed in a paragraph. The simulated data have the advantage that the ideal solution is known and thus a doubtless evaluation is possible. (Discussion will be added on page 14)

*Regarding the test exercise of Section 5, Table 1 shows that the largest errors are associated with g21 and, to some extent, with h21 . Is this by chance, or related to some systematics?*

**Reply:** The deviation of these coefficients is related to the underlying model and therefore it is systematic noise. (Discusson will be added on page 14.

*3. Technicalities. Considering the target audience of the journal, different to a good extent from the signal processing community, the mathematical language of the paper may prevent the optimal transmission of the message. Additional explanations may help, inserted in the text or collected in an Appendix – when detailing the math would perturb the flow too much:*

*• L69-71: Please clarify this sentence, possibly including an example.*
**Reply:** For example, when the magnetic field data are known on a dense grid in the vicinity of the planet, the Gauss coefficients can be estimated via integration. But in the case of a limited data set those integrals cannot be evaluated. (Sentence to be added on page 3.)

*• L93-97: The intuitive introduction of the filter matrix w via Eq. 8 s a bit confusing, since eventually the non-parametrized (external) part of B does not show up in the g C formula, Eq. 41.*
**Reply:** Eq. 8 has been written down, since it is the first intuitive idea to solve the inverse problem. An explanation that the non-parameterized parts <v> are unknown and have to be truncated by w will be added on page 4.

*• Eqs. 9 and 10 fall pretty much out of the blue. The use of M and P becomes clear later, but some clarification would be good already at this point.*
**Reply:** Agreed. We will add motivation on page 4.

*• L106-110: Please detail why the determinant vanishes (even though it may look straight), how does statistical average prevent this, how is statistical average achieved.*

**Reply:** It is a mathematical nature. The calculation of vanishing determinant of the outer product will be shown in Appendix.

• *L127: Please indicate also the second order moments.*
**Reply:** Maybe the structure of the sentence was confusing, but <g o g> are the second order moments.

• *Eq. 21: Please explain why 2<Hg> o <v> and not <Hg> o <v> + <v> o <Hg> (given that, in general, the external product does not commute).*
**Reply:** The outer product commutates in this special case because <Hg> and <v> have the same dimension (to be added on page 6)

• *Eq. 27 and L154: Please explain why this is not enough to uniquely determine w.*
**Reply:** It is because the parts that have to be truncated are unknown. (will be added on page 7)

• *L155-158: Feels confuse. As long as the filter matrix truncates the non-parametrized part, it is not clear why its contribution to the data matters, neither how 'this yields the following procedure'.*
**Reply:** The contribution matters because the parts that have to be truncated are unknown. The sentence 'this yields the following procedure' will be deleted.

• *L191: Why is tr P a convex function?*
**Reply:** trP is a convex function since it is the sum of the quadratic averaged expansion coefficients and thus it follows the same structure as for example the function $f(x)=x^2$ (explanation to be added on p. 8)

•*Eqs. 42 and 43: Please detail what is meant by 'input' and 'output'. Regarding Eq. 43, is there an equation analogous to Eq. 23, to clarify the meaning of 'signal' and 'noise' also for output?*
**Reply:** By input we mean the measured data. By output we mean the filtered data (will be added on p. 9);
Yes, one can construct an equation analogous to Eq. 23, which results, when $w^T(...)w$ is applied to this equation. This can be seen within SNR_o.

•*Eq. 44: Please explain why this ratio is dominated by 1/trace.*
**Reply:** The array gain is dominated by 1/trace, since P, H and v are given by the model and the data and do not depend on the method (w does). (explanation will be added on page 9)

•*L266-267: Please provide a brief demonstration.*
**Reply:** The eigenvalues of M will be calculated in the Appendix.

•*L267-269: Please explain briefly what is this about.*
**Reply:** Explanation will be added on page 11.

•*L278: How is the 'compromise' quantified?*
**Reply:** The compromise can be understood in the sense that trace reaches its minimal value under the condition that P_d is maximal. (to be added on p. 11)

•*L278-282: This is quite opaque for those not familiar with signal processing and in particular with these techniques.*
**Reply:** The basic idea of the Tikhonov regularisation will briefly be explained on pages 11–12.

*4. Others*
• *L17-18: What is non-ideal orbits?*
**Reply:** The sentence was adversely formulated. We change it to „data sampled on single orbits".

• *L18: simulated Mercury magnetic field data*
**Reply:** Formulation will be changed in the manuscript.

• *L54: 'closing the void' => 'covering the range' ?*
**Reply:** Formulation will be changed in the manuscript.

• *Eq. 54 is identical to Eq. 41.*
**Reply:** Reference will be added in the manuscript.

• *L325: in => at*
**Reply:** Formulation will be changed in the manuscript.

• *L352: In principle, one could analyze also the external field, if some model is adopted.*
**Reply:** That's very true! We will modify the sentence accordingly.

---

## Referee Comment (RC2) · Anonymous Referee #2 · 25 Oct 2020

The manuscript presents well founded support for mathematical analysis of planetary magnetic field basing on experimental data.

1. Line 33: "N data points xi, i = 1, ...,N" and Line 102 "Q indicates the number of measurements". Are N and Q the same numbers? 2. Line 94: "fulfills in principle the resolution of Eq. (7) with respect to g." Did you mean solution of Eq.(7)? 3. As a usual practice, for validation of the model, experimental data are divided into two parts. The first one is used for selection/tuning of model parameters with the help of the various optimization algorithms. The second part provides verification of the model by means comparison of experimental data with the data predicted by the built model. It could be helpful to demonstrate such an approach here.

---

## Author Comment (AC2) · 26 Oct 2020

*The manuscript presents well founded support for mathematical analysis of planetary magnetic field basing on experimental data. 1. Line 33: "N data points xi, i = 1, …,N" and Line 102 "Q indicates the number of measurements". Are N and Q the same numbers? 2. Line 94: "fulfills in principle the resolution of Eq. (7) with respect to g." Did you mean solution of Eq.(7)? 3. As a usual practice, for validation of the model, experimental data are divided into two parts. The first one is used for selection/tuning of model parameters with the help of the various optimization algorithms. The second part provides verification of the model by means comparison of experimental data with the data predicted by the built model. It could be helpful to demonstrate such an approach here.*

**Reply:**
**1.)** Q and N are not the same numbers. N is the number of spatial data points, whereas Q indicates the number of measurements at each of these data points (for example the number of flybys at each point). A comment will be added in the manuscript.

**2.)** Agreed, we will modify the word „resolution" to „solution".

**3.)** For the application of several inversion methods (e.g. machine learning) it is useful/necessary to devide the data into two parts. Capon's method does not require this segmentation. For example, each data set corresponds with a diagonal loading parameter. Since this parameter depends on the measurements and on the underlying model, it has to be calculated for each data set individually. When the data and the model are known, for each data set the diagonal loading parameter is calculated with the measurements and the data points itself and then Capon's estimator can be calculated directly.

---

## Author Response (AR1)

**Reviewer 1:**

*The paper Mathematical foundation of Capon's method for planetary magnetic field analysis provides the underlying formalism for applying Capon's method to planetary magnetic fields and illustrates it with simulated data relevant for BepiColombo mission. While this is a valuable contribution to the field, a number of points need to be addressed before publication.*

*1. Extension of Capon's method to planetary magnetic fields. As indicated in the first and last paras of Section 3, L73–75 and L202–206, the paper generalizes the Capon method, previously used for the analysis of wave data. This is a major result that could be emphasized better, perhaps in a separate discussion section. This section could include a closer analysis of the case in the paper as compared to the wave case, by referring to Motschmann et al. (1996).*
*The discussion section could also detail the key principle(s) underlying the method, like maximum likelihood / minimum variance, along the line of Narita (2019). The divergence free feature of the magnetic field could be discussed on top, as in Motschmann et al. (1996).*

**Reply:** Agreed. We added a section dedicated to the connection of the Capon method to other inversion methods (e.g., maximum likelihood, least square fit) and highlighted the difference from the wave analysis method (Motschmann et al., 1996) on p. 16. We also added a discussion about the divergence-free nature of  the magnetic field (p. 2, ll. 45--52).

*Related:*
*• L166-168: I am not sure I understand the text here, even though I could essentially follow Eqs. 30 to 41. Eventually, the underlying model makes the main contribution to the data (e.g., in the test application of Section 5). I guess this 'minimal contribution' has rather the meaning of Motschmann et al. (1996), where the filter w absorbs all the energy not associated with k (here not associated with the parametrized field) and leaves the part related to k undistorted (here the part related to the parametrized field). Same issue at L256.*

**Reply:** Surely it is possible that the underlying model makes the main contribution to the data. But the distribution of the parameterized and the non-parameterized parts is unknown. Therefore, we stay conservative and assume safely that a large part of data is influenced by the noise and non-parametrized signals. (Explanation added on p. 7, l. 182 f)

*• L200-201: Is this expression derived by Narita (2019)? Or could be derived by further processing of the maximum likelihood estimator? (e.g., in the suggested discussion section?)*

**Reply:** Yes, Narita (2019) derived the expression for Capon's estimator by regarding the likelhood function as nearly Gaussian (particularly around the peak of the likelhood function).

*2. Illustration & Validation. In view of upcoming BepiColombo data, the authors chose to illustrate the method with simulated observations of Mercury magnetic field. While this is certainly helpful to prepare BepiColombo, I wonder if it is also the best test bed for the method. Earth magnetic field is known much better, at various altitudes – such that the weight of the external field and its influence on the results could be analyzed too. Including an example at the Earth, or at least a brief discussion of this validation possibility, would be more than welcomed.*

**Reply:** The application of Capon's method to the analysis of Mercury's magnetic field has been emphasized as an example. Surely, the test against the Earth' magnetic field would be a good alternative. But such an analysis would have the size of a separate paper and it could not be pressed in a paragraph. The simulated data have the advantage that the ideal solution is known and thus a doubtless evaluation is possible. (Discussion added on p. 15, ll. 336--339)

*Regarding the test exercise of Section 5, Table 1 shows that the largest errors are associated with g21 and, to some extent, with h21 . Is this by chance, or related to some systematics?*

**Reply:** The deviation of these coefficients is related to the underlying model and therefore it is systematic noise. (Discusson added on p. 15, ll. 352--354).

*3. Technicalities. Considering the target audience of the journal, different to a good extent from the signal processing community, the mathematical language of the paper may prevent the optimal transmission of the message. Additional explanations may help, inserted in the text or collected in an Appendix – when detailing the math would perturb the flow too much:*

*• L69-71: Please clarify this sentence, possibly including an example.*
**Reply:** For example, when the magnetic field data are known on a dense grid in the vicinity of the planet, the Gauss coefficients can be estimated via integration. But in the case of a limited data set those integrals cannot be evaluated. (Sentence added on p. 3, ll. 78--80)

*• L93-97: The intuitive introduction of the filter matrix w via Eq. 8 s a bit confusing, since eventually the non-parametrized (external) part of B does not show up in the g C formula, Eq. 41.*
**Reply:** Eq. 8 has been written down, since it is the first intuitive idea to solve the inverse problem. An explanation that the non-parameterized parts <v> are unknown and have to be truncated by w has been added on p. 4, l. 104.

*• Eqs. 9 and 10 fall pretty much out of the blue. The use of M and P becomes clear later, but some clarification would be good already at this point.*
**Reply:** Agreed. We added motivation on page 4, l. 109.

*• L106-110: Please detail why the determinant vanishes (even though it may look straight), how does statistical average prevent this, how is statistical average achieved.*
**Reply:** It is a mathematical nature. The calculation of vanishing determinant of the outer product is shown in Appendix A on page 18. The statistical average is achieved by averaging over several numbers of measurements (added on p. 5, l. 119)

*• L127: Please indicate also the second order moments.*
**Reply:** Maybe the structure of the sentence was confusing, but <g o g> are the second order moments. We changed the order of the sentence and the equation accordingly (p. 6, l. 140).

*• Eq. 21: Please explain why 2<Hg> o <v> and not <Hg> o <v> + <v> o <Hg> (given that, in general, the external product does not commute).*
**Reply:** The outer product commutates in this special case because <Hg> and <v> have the same dimension (added on p. 6, l. 156).

*• Eq. 27 and L154: Please explain why this is not enough to uniquely determine w.*
**Reply:** It is because the parts that have to be truncated are unknown. (added on p. 7, l. 174)

*• L155-158: Feels confuse. As long as the filter matrix truncates the non-parametrized part, it is not clear why its contribution to the data matters, neither how 'this yields the following procedure'.*
**Reply:** The contribution matters because the parts that have to be truncated are unknown. The sentence 'this yields the following procedure' has been deleted (p. 7, l. 173).

*• L191: Why is tr P a convex function?*
**Reply:** trP is a convex function since it is the sum of the quadratic averaged expansion coefficients and thus it follows the same structure as for example the function $f(x)=x^2$ (explanation added on p. 8, Eq. 41)

*•Eqs. 42 and 43: Please detail what is meant by 'input' and 'output'. Regarding Eq. 43, is there an equation analogous to Eq. 23, to clarify the meaning of 'signal' and 'noise' also for output?*
**Reply:** By input we mean the measured data. By output we mean the filtered data (added on p. 9, l. 227 and l. 230);
Yes, one can construct an equation analogous to Eq. 23, which results, when $w^T(...)w$ is applied to this equation. This can be seen within SNR_o.

*•Eq. 44: Please explain why this ratio is dominated by 1/trace.*
**Reply:** The array gain is dominated by 1/trace, since P, H and v are given by the model and the data and do not depend on the method (w does). (explanation added on page 9, l. 235)

•*L266-267: Please provide a brief demonstration.*
**Reply:** The eigenvalues of M are calculated in the Appendix B (p. 18/19).

•*L267-269: Please explain briefly what is this about.*
**Reply:** Done. (p. 12, l. 288)

•*L278: How is the 'compromise' quantified?*
**Reply:** The compromise can be understood in the sense that trace reaches its minimal value under the condition that P_d is maximal. (added on p. 12, l.297/298)

•*L278-282: This is quite opaque for those not familiar with signal processing and in particular with these techniques.*
**Reply:** The basic idea of the Tikhonov regularisation is briefly explained on page 12, ll. 299--304.

*4. Others*
• *L17-18: What is non-ideal orbits?*
**Reply:** The sentence was adversely formulated. We changed it to „data sampled on single orbits" (p. 1, l. 18).

• *L18: simulated Mercury magnetic field data*
**Reply:** Formulation has been changed (p. 1, l. 18).

• *L54: 'closing the void' => 'covering the range' ?*
**Reply:** Done. (p. 3, l. 60)

• *Eq. 54 is identical to Eq. 41.*
**Reply:** Reference has been added. (p. 14, l. 312)

• *L325: in => at*
**Reply:** Done. (p. 15, l. 355)

• *L352: In principle, one could analyze also the external field, if some model is adopted.*
**Reply:** That's very true! Sentence has been added. (p. 15, l. 356)

**Reviewer 2:**

*The manuscript presents well founded support for mathematical analysis of planetary magnetic field basing on experimental data. 1. Line 33: "N data points xi, i = 1, ...,N" and Line 102 "Q indicates the number of measurements". Are N and Q the same numbers? 2. Line 94: "fulfills in principle the resolution of Eq. (7) with respect to g." Did you mean solution of Eq.(7)? 3. As a usual practice, for validation of the model, experimental data are divided into two parts. The first one is used for selection/tuning of model parameters with the help of the various optimization algorithms. The second part provides verification of the model by means comparison of experimental data with the data predicted by the built model. It could be helpful to demonstrate such an approach here.*

**Reply:**
1.) Q and N are not the same numbers. N is the number of spatial data points, whereas Q indicates the number of measurements at each of these data points (for example the number of flybys at each point). A comment has been added on p. 5, l. 113.

2.) Agreed, we modified the word „resolution" to „solution" (p. 4, l. 103).

3.) For the application of several inversion methods (e.g. machine learning) it is useful/necessary to divide the data into two parts. Capon's method does not require this segmentation. For example, each data set corresponds with an optimal diagonal loading parameter. Since this parameter depends on the measurements and on the underlying model, it has to be calculated for each data set individually. When the data and the model are known, for each data set the diagonal loading parameter is calculated with the measurements and the data points themselves and then Capon's estimator can be calculated directly.

**General changes in the manuscript:**

- Changes in the manuscript are marked with „latexdiff", i.e., added text is marked in blue and the old version of the formulation is crossed out and marked in red
- The position of changes that are related to Reviewer comments are directly stated at the reply.
- We added a section about the discussion of Capon's method on p. 16.
- Appendix A and B have been added on p. 18/19
- The literature list has been extended by:
  Tikhonov, A. N., Goncharsky, A., Stepanov, V. V., Yagola, A. G.: Numerical Methods for the Solution of Ill-Posed Problems, Springer Netherlands, 1995. ISBN 079233583X

[revised manuscript text omitted]

where $\boldsymbol{n} = \left(\boldsymbol{n}^1 \ldots \boldsymbol{n}^N\right)^T \in \mathbb{R}^{3N}$, $\boldsymbol{v} = \left(\boldsymbol{v}^1 \ldots \boldsymbol{v}^N\right)^T \in \mathbb{R}^{3N}$ and $\mathbf{H} = \left[\mathbf{H}^1 \ldots \mathbf{H}^N\right]^T \in \mathbb{R}^{3N \times G}$. $G$ indicates the number of expansion coefficients.

Within Eq. (6) the magnetic field vector $\boldsymbol{B}$ and the shape matrix $\mathbf{H}$, given by the underlying model, are known. The coefficient vector $\boldsymbol{g}$ is to be determined by data fitting. Since in most applications the number of known magnetic data points is much larger than the number of wanted expansion coefficients ($G \ll 3N$), $\mathbf{H}$ is a rectangular matrix in general. Furthermore, the non-parameterized parts of the field and the noise are unknown. Therefore, the direct inversion of Eq. (6) is impossible and $\boldsymbol{g}$ has to be estimated. In this case, Capon's method establishes a robust and useful tool to find the estimated solution for the expansion coefficients in Eq. (6).

Since the method does not require the orthogonality of the basis functions, it has a wider range of applications when decomposing the measured data into a set of superposed signals, especially when the number of data points is limited. For example, when the magnetic field data are measured on a dense grid in the vicinity of the planet, the Gauss coefficients can be estimated via integration of the data. But in the case of a limited data set those integrals cannot be evaluated.

**3 Derivation of Capon's method**

[revised manuscript text omitted]

with its related error

$$\langle \boldsymbol{\varepsilon} \rangle = \frac{1}{Q} \sum_{\alpha=1}^{Q} \boldsymbol{\varepsilon}^\alpha \tag{16}$$

is available.

In contrast to the estimator, the true coefficient vector is a theoretical given vector, that is not affected by the averaging ($\boldsymbol{g} \equiv \langle \boldsymbol{g} \rangle$, Eq. (9)). This property directly links the estimator to the true coefficient vector, which can be rewritten as

$$\boldsymbol{g} = \boldsymbol{g}_C^\alpha + \boldsymbol{\varepsilon}^\alpha. \tag{17}$$

Averaging over $Q$ measurements and using $\boldsymbol{g} \equiv \langle \boldsymbol{g} \rangle$ results in

$$\boldsymbol{g} = \boldsymbol{g}_C + \langle \boldsymbol{\varepsilon} \rangle \tag{18}$$

and

$$\boldsymbol{g} \circ \boldsymbol{g} = \langle \boldsymbol{g}_C \circ \boldsymbol{g}_C \rangle + \langle \boldsymbol{g}_C \circ \boldsymbol{\varepsilon} \rangle + \langle \boldsymbol{\varepsilon} \circ \boldsymbol{g}_C \rangle + \langle \boldsymbol{\varepsilon} \circ \boldsymbol{\varepsilon} \rangle$$

140 analogously for the second order moments .

$$\boldsymbol{g} \circ \boldsymbol{g} = \langle \boldsymbol{g}_C \circ \boldsymbol{g}_C \rangle + \langle \boldsymbol{g}_C \circ \boldsymbol{\varepsilon} \rangle + \langle \boldsymbol{\varepsilon} \circ \boldsymbol{g}_C \rangle + \langle \boldsymbol{\varepsilon} \circ \boldsymbol{\varepsilon} \rangle. \tag{19}$$

In the limit of vanishing errors $\langle \boldsymbol{\varepsilon} \rangle \to 0$, $\langle \boldsymbol{\varepsilon} \circ \boldsymbol{\varepsilon} \rangle \to 0$ respectively, Capon's estimator converges to the true coefficient vector

$$\boldsymbol{g}_C \to \boldsymbol{g} \tag{20}$$

and therefore

[revised manuscript text omitted]

The deviation $\left|\boldsymbol{g}_C - \boldsymbol{g}\right|$ between Capon's estimator and the true coefficient vector results in $30.2\,\mathrm{nT}$ or $14.7\,\%$, respectively, for the optimal diagonal loading parameter. When the magnetic field data are evaluated at an ensemble of data points with a distance of $0.2\,R_\mathrm{M}$ from Mercury's surface this deviation is of the same order (Toepfer et al., 2020). Since the underlying model neglects the external parts and only the internal parts are considered, the coefficients $g_2^1$ and $h_2^1$ show large deviations to the implemented coefficients. Extending the underlying model by a parameterization of the external parts of the magnetic field improves Capon's estimator especially when the magnetic field data are evaluated  at some distance above Mercury's surface. In principle, one can analyze also the external field, if some model is adopted.

**6 Discussion of Capon's method**

Capon's method has formerly been applied to the analysis of waves. Thus, the existing derivations treat the non-parameterized parts of the field $v$ as Gaussian noise, so that $v$ and the measurement noise $n$ are of the same character (Motschmann et al., 1996). Considering the analysis of planetary magnetic fields this assumption is indefensible. For example, the external parts of the field are systematic noise and cannot be modeled by a Gaussian distribution. When the non-parameterized parts are Gaussian, i.e. $\langle v \rangle = 0$, the truncation of these parts by the filter matrix $\mathbf{w}^\dagger \langle v \rangle = 0$ (cf. Eq. 27) is fulfilled trivially, which reduces the terms within the derivation and the estimation of the diagonal loading parameter. Therefore, the above presented mathematical foundations generalize the previous derivations of Capon's method and transit into the derivation presented by Motschmann et al. (1996) for the special case of $\langle v \rangle = 0$.

As already mentioned in the introduction (Sec. 1), Capon's method can be regarded from several mathematical perspectives. Within the derivation presented above, the output power $P$, which is defined as the trace of the coefficient (covariance) matrix is minimized with respect to the filter matrix $\mathbf{w}$, subject to the distortionless contraint $\mathbf{w}^\dagger \mathbf{H} = \mathbf{I}$. This procedure corresponds with the name Minimum Variance Distortionless Response Estimator (MVDR), since $P$ contains the variance of the model coefficients. Narita (2019) showed, that Capon's estimator can also be derived by treating Capon's method as a special case of the maximum likelihood estimator by regarding the likehood function as nearly Gaussian (particularly around the peak of the likelihood function). As discussed in Sec. 5, Capon's method can also be interpreted as a special case of the weighted least square fit. This illustrates, that the several existing inversion methods for linear inverse problems are connected with each other and are not as different as they seem to be at first appearance.

**7 Conclusions**

Capon's method is a robust and useful tool for various kinds of ill-posed inverse problems, such as Mercury's planetary magnetic field analysis. The derivation of the method can be regarded from different mathematical perspectives. Here we revisited the linear-algebraic matrix formulation of the method and extended the derivation for Mercury's magnetic field analysis. Capon's method becomes even more robust by incorporating the diagonal loading technique. Thereby, the construction of a filter matrix is vital to the derivation of Capon's estimator.

Especially the trace of the filter matrix determines the array gain, which is defined as the ratio of the output and the input signal-noise-ratio. If the trace is large, the output signal-noise-ratio can decrease resulting in signal elimination and thus, the performance of Capon's estimator degrades. Bounding the trace of the filter matrix results in diagonal loading of the data covariance matrix, which improves the robustness of the method. The main problem of the diagonal loading technique is that in general it is not clear how to choose the optimal diagonal loading parameter. Since for the analysis of planetary magnetic fields we prefer measurements that are stationary up to the Gaussian noise, estimators for the diagonal loading parameter, that are related to eigenvalues of the data covariance matrix corresponding to interference and noise cannot be applied. Making use of the L-curve's technique enables a robust procedure for estimating the optimal diagonal loading parameter.

For the calculation of Capon's estimator several computationally burdensome matrix inversions are necessary. Interpretation of Capon's method as a special case of the least square fit method enables the usage of numerically more stable and less burden minimization algorithms, e.g. gradient descent or conjugate gradient method, for calculating Capon's estimator.

It should be noted that the parameterization of Mercury's internal magnetic field via the Gauss representation, as mentioned in section 2, is only one of several possibilities for modeling the magnetic field in the vicinity of Mercury. The underlying model can be extended by other parameterizations, for example the Mie representation (toroidal-poloidal decomposition) or magnetospheric models and Capon's method can be applied to estimate the related model coefficients. Besides the analysis of Mercury's internal magnetic field, the extention of the model also enables the reconstruction of current systems flowing in the magnetosphere. Concerning the BepiColombo mission this work establishes a mathematical basis for the application of Capon's method to analyze Mercury's internal magnetic field in a robust and manageable way.

*Data availability.* The raw data supporting the conclusions of this article will be made available by the authors, without undue reservation.

**Appendix A: Determinant of the outer product**

405 The influence of the averaging to the determinant of the data covariance marix $\mathbf{M}$ is exemplarily illustrated for the three-dimensional case. Thus, the magnetic field vector is given by

$$\boldsymbol{B} = \begin{pmatrix} B_x \\ B_y \\ B_z \end{pmatrix} \tag{A1}$$

and therefore, the outer product results in

$$\boldsymbol{B} \circ \boldsymbol{B} = \begin{bmatrix} B_x^2 & B_x B_y & B_x B_z \\ B_x B_y & B_y^2 & B_y B_z \\ B_x B_z & B_y B_z & B_z^2 \end{bmatrix} \tag{A2}$$

410 with a vanishing determinant

$$\det(\boldsymbol{B} \circ \boldsymbol{B}) = 3 B_x^2 B_y^2 B_z^2 - 3 B_x^2 B_y^2 B_z^2 = 0. \tag{A3}$$

Throughout the averaging of the data, the data covariance matrix results in

$$\mathbf{M} = \langle \boldsymbol{B} \circ \boldsymbol{B} \rangle = \langle \boldsymbol{B} \rangle \circ \langle \boldsymbol{B} \rangle + \sigma_n^2 \underline{\underline{I}} = \begin{bmatrix} \langle B_x^2 \rangle + \sigma_n^2 & \langle B_x \rangle \langle B_y \rangle & \langle B_x \rangle \langle B_z \rangle \\ \langle B_x \rangle \langle B_y \rangle & \langle B_y^2 \rangle + \sigma_n^2 & \langle B_y \rangle \langle B_z \rangle \\ \langle B_x \rangle \langle B_z \rangle & \langle B_y \rangle \langle B_z \rangle & \langle B_z^2 \rangle + \sigma_n^2 \end{bmatrix} \tag{A4}$$

with a non-vanishing determintant

415 $\det(\mathbf{M}) = (\langle B_x^2 \rangle + \sigma_n^2)(\langle B_y^2 \rangle + \sigma_n^2)(\langle B_z^2 \rangle + \sigma_n^2) - \langle B_x^2 \rangle \langle B_y^2 \rangle \langle B_z^2 \rangle - \sigma_n^2 \langle B_y^2 \rangle \langle B_z^2 \rangle - \sigma_n^2 \langle B_x^2 \rangle \langle B_z^2 \rangle - \sigma_n^2 \langle B_x^2 \rangle \langle B_y^2 \rangle$ (A5)

$$= \sigma_n^4 (\langle B_x^2 \rangle + \langle B_y^2 \rangle + \langle B_z^2 \rangle) + \sigma_n^6 \tag{A6}$$

$$\neq 0. \tag{A7}$$

Thus, the inverse of $\mathbf{M}$ exists, whereas the outer product $\boldsymbol{B} \circ \boldsymbol{B}$ is singular.

**Appendix B: Eigenvalues of the data covariance matrix**

420 The data covariance matrix is defined as

$$\mathbf{M} = \langle \boldsymbol{B} \circ \boldsymbol{B} \rangle = \langle \boldsymbol{B} \rangle \circ \langle \boldsymbol{B} \rangle + \sigma_n^2 \mathbf{I}. \tag{B1}$$

This matrix is quadratic and especially diagonalisable. Thus, there exists a matrix $\mathbf{D_M}$ which is similar to the matrix $\mathbf{M}$, so that

$$\mathbf{D_M} = \mathbf{V}^\dagger \mathbf{M} \mathbf{V}, \tag{B2}$$

where $\mathbf{V}$ is an orthogonal transformation, i.e. $\mathbf{V}^\dagger \mathbf{V} = \mathbf{I}$ and $\mathbf{D_M}$ is a diagonal matrix which diagonal elements are given by the eigenvalues of $\mathbf{M}$. Inserting the definition of the matrix $\mathbf{M}$ delivers

$$\mathbf{D_M} = \mathbf{V}^\dagger \mathbf{M} \mathbf{V} = \mathbf{V}^\dagger \langle \boldsymbol{B} \circ \boldsymbol{B} \rangle \mathbf{V} = \mathbf{V}^\dagger \left( \langle \boldsymbol{B} \rangle \circ \langle \boldsymbol{B} \rangle \right) \mathbf{V} + \sigma_n^2 \mathbf{I}, \tag{B3}$$

since $\mathbf{V}^\dagger \mathbf{V} = \mathbf{I}$. To determine the diagonal form of the outer product, the two-dimensional case where

$$\langle \boldsymbol{B} \rangle = \begin{pmatrix} \langle B_x \rangle \\ \langle B_y \rangle \end{pmatrix} \tag{B4}$$

is considered exemplarily. Thus,

$$\langle \boldsymbol{B} \rangle \circ \langle \boldsymbol{B} \rangle = \begin{bmatrix} \langle B_x \rangle^2 & \langle B_x \rangle \langle B_y \rangle \\ \langle B_x \rangle \langle B_y \rangle & \langle B_y \rangle^2 \end{bmatrix} \tag{B5}$$

and the characteristic polynomial results in

$$\left( \langle B_x \rangle^2 - \beta \right) \left( \langle B_y \rangle^2 - \beta \right) - \langle B_x \rangle^2 \langle B_y \rangle^2 = 0 \tag{B6}$$

or equivalently

$$\beta^2 - \beta \left( \langle B_x \rangle^2 + \langle B_y \rangle^2 \right) = \beta^2 - \beta \left| \langle \boldsymbol{B} \rangle \right|^2 = 0, \tag{B7}$$

where $\beta$ denotes the eigenvalue which is given by $\beta = 0$ and $\beta = \left| \langle \boldsymbol{B} \rangle \right|^2$. Therefore, in general the diagonal matrix of the outer product is given by

$$\mathbf{D}_{\langle \boldsymbol{B} \rangle \circ \langle \boldsymbol{B} \rangle} = \begin{bmatrix} \left| \langle \boldsymbol{B} \rangle \right|^2 & 0 & \cdots & 0 \\ 0 & 0 & \cdots & 0 \\ \vdots & & \ddots & \vdots \\ 0 & \cdots & \cdots & 0 \end{bmatrix}. \tag{B8}$$

The noise matrix $\sigma_n^2 \mathbf{I}$ is already a diagonal matrix, so that the diagonal form of $\mathbf{M}$ is given by

$$\mathbf{D_M} = \begin{bmatrix} \left| \langle \boldsymbol{B} \rangle \right|^2 & 0 & \cdots & 0 \\ 0 & 0 & \cdots & 0 \\ \vdots & & \ddots & \vdots \\ 0 & \cdots & \cdots & 0 \end{bmatrix} + \sigma_n^2 \begin{bmatrix} 1 & 0 & \cdots & 0 \\ 0 & 1 & \cdots & 0 \\ \vdots & & \ddots & \vdots \\ 0 & \cdots & \cdots & 1 \end{bmatrix} = \begin{bmatrix} \left| \langle \boldsymbol{B} \rangle \right|^2 + \sigma_n^2 & 0 & \cdots & 0 \\ 0 & \sigma_n^2 & \cdots & 0 \\ \vdots & & \ddots & \vdots \\ 0 & \cdots & \cdots & \sigma_n^2 \end{bmatrix}. \tag{B9}$$

[revised manuscript text omitted]